# HDT: Hierarchical Document Transformer

**Haoyu He**[1,2][*] **Markus Flicke**[1,2][*], **Jan Buchmann**[3], **Iryna Gurevych**[3], **Andreas Geiger**[1,2]
[1] University of Tübingen      [2] Tübingen AI Center
[3] Technical University of Darmstadt and Hessian Center for AI (hessian.AI)
{haoyu.he,markus.flicke,a.geiger}@uni-tuebingen.de
{jan.buchmann,iryna.gurevych}@tu-darmstadt.de

## Abstract

In this paper, we propose the Hierarchical Document Transformer (HDT), a novel sparse Transformer architecture tailored for structured hierarchical documents. Such documents are extremely important in numerous domains, including science, law or medicine. However, most existing solutions are inefficient and fail to make use of the structure inherent to documents. HDT exploits document structure by introducing auxiliary anchor tokens and redesigning the attention mechanism into a sparse multi-level hierarchy. This approach facilitates information exchange between tokens at different levels while maintaining sparsity, thereby enhancing computational and memory efficiency while exploiting the document structure as an inductive bias. We address the technical challenge of implementing HDT's sample-dependent hierarchical attention pattern by developing a novel sparse attention kernel that considers the hierarchical structure of documents. As demonstrated by our experiments, utilizing structural information present in documents leads to faster convergence, higher sample efficiency and better performance on downstream tasks.

## 1 Introduction

Many natural language processing tasks including summarization and question answering require language models to encode long structured documents such as scientific papers or Wikipedia articles into meaningful representations. Since 2017, attention-based Transformer architectures Vaswani et al. (2017) have established themselves as the dominant modeling paradigm, demonstrating state-of-the-art performance on numerous NLP tasks. While dense attention yields context-rich representations, its computational complexity grows quadratically with the input. This is problematic when processing large documents, in particular on resource constrained hardware such as consumer GPUs.

At the same time, most documents are naturally structured: Words form sentences, sentences form sections and sections form documents. Surprisingly, this structure is largely ignored by many existing language models that typically consider their context as a "flat" sequence of tokens. While transformer models can in principle learn to generalize hierarchically, *structural grokking* requires extremely long training (Murty et al., 2023). Moreover, the performance of flat long context models depends on the position of relevant information (Liu et al., 2023). We hypothesize that exploiting the structure of documents explicitly yields two major benefits: (1) Imposing the structure of documents as an inductive bias improves sample efficiency and generalization. (2) Adapting the attention pattern to the document structure leads to sparse representations which reduce computational and memory complexity, and enable processing of long documents even on consumer hardware.

To test this hypothesis, we propose the *Hierarchical Document Transformer (HDT)*, a novel sparse Transformer architecture for processing hierarchically structured documents. More specifically, we first introduce auxiliary anchor tokens for all structural elements such as sentences, sections and documents as illustrated in Fig. 1a. Second, we redesign the

---

[*]Equal contribution.

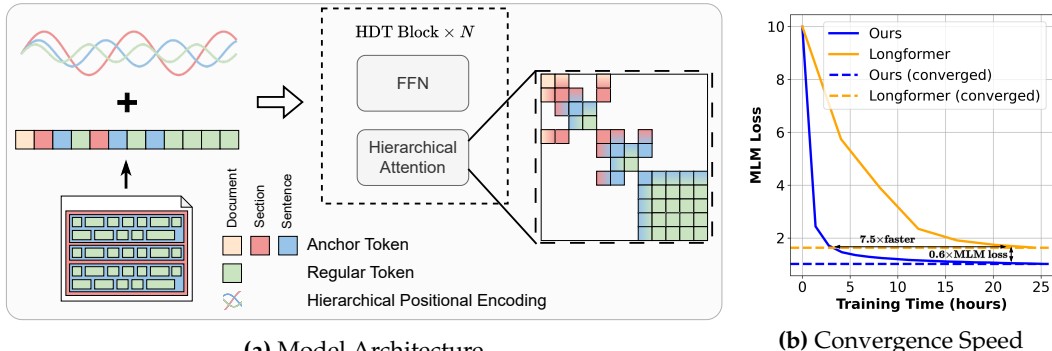

**(a)** Model Architecture      **(b)** Convergence Speed

**Figure 1:** (a) We propose a sparse attention kernel that considers the hierarchical structure of documents. Here, regular tokens are illustrated in green, and auxiliary anchor tokens in yellow (document), red (section) and blue (sentence). Each token attends to its parent, siblings and children. Cross-level attention is illustrated using color gradients in the attention matrix. Utilizing structural information present in documents leads to faster pre-training (b) and better performance on downstream tasks. We use the held-out validation set in (b) to calculate the MLM loss.

attention pattern of a Transformer block into a multi-level hierarchy where information is exchanged only between tokens at the same level (siblings) as well as between the respective parent and child tokens. By stacking multiple HDT blocks, information from any token can reach any other token. At the same time, the attention pattern of HDT blocks is highly sparse, leading to gains in computational and memory efficiency. In contrast to previous hierarchical models Wu et al. (2021a); Chalkidis et al. (2022), our HDT block supports deeper hierarchies and communicates at all hierarchy levels *simultaneously* such that stacking of only a few HDT layers establishes communication between all tokens.

However, implementing our model required solving a key technical challenge: Existing sparse Transformer architectures like LongFormer Beltagy et al. (2020) assume a fixed sparsity pattern for all inputs in a mini-batch which can be implemented using standard libraries. In contrast, HDT imposes a different sparsity pattern for each sample within a mini-batch as each document is structured differently. Towards this goal, we developed a novel, flexible and efficient attention kernel based on the Triton (Tillet et al., 2019) library. We pre-train HDT encoder models using Masked Language Modeling (MLM) and HDT encoder-decoder models using UL2 (Tay et al., 2023b) on three document datasets: arXiv (Saier et al., 2023), HUPD Patents (Suzgun et al., 2023) and Wikipedia. We demonstrate improved pre-training convergence rates (see Fig. 1b) as well as better downstream task performance on several proximity, summarization, QA and NLI tasks. Our code and data are available at https://github.com/autonomousvision/hdt.

## 2   Related Work

This study combines two lines of work that try to improve the performance of bi-directional Transformers in handling long-document inputs: *structural modeling* and *efficient attention*.

Structural modeling approaches exploit the hierarchical organization of documents into sections, subsections, paragraphs, sentences, etc. They can be divided into two categories Buchmann et al. (2024): (1) In *structure infusion*, structural information is added to the Transformer input, e.g., via special tokens (Aghajanyan et al., 2022; Liu et al., 2022; Buchmann et al., 2024), position embeddings (Bai et al., 2021; Cao & Wang, 2022), attention masks (Wang et al., 2019; Liu et al., 2021; Hong et al., 2022; Wu et al., 2021b) or fusing self-attention layers with GNN layers Sachan et al. (2021); Ahmed et al. (2019). While observing performance improvements, these works do not improve efficiency. (2) *Hierarchical processing* employs architectures that first contextualize tokens on a local (e.g., sentence) level, aggregate the local representations (e.g., in a "[CLS]" token) and then contextualize the aggregates on one or several higher levels (e.g., paragraph or section, Yang et al. 2016; Chalkidis et al. 2019; Yang 2019; Ruan et al. 2022; Zhang et al. 2022; Dai et al. 2022). Local and higher-level

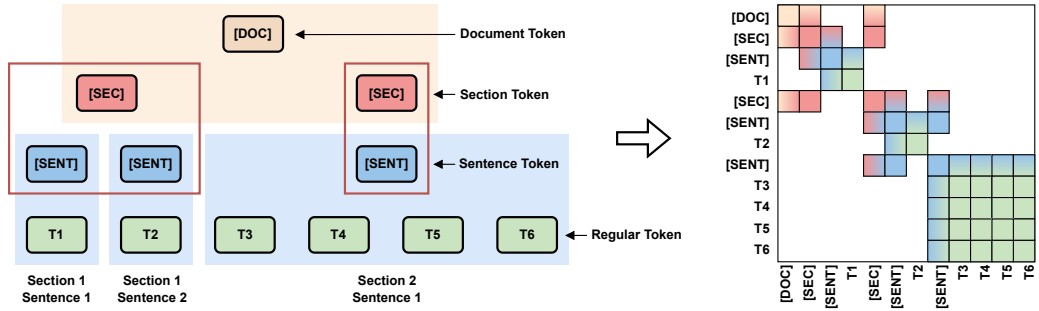

**Figure 2: Hierarchical Document Decomposition.** Left: Tree representation of a document. Tokens within the same box attend to each other. Tokens that do not share a box attend to each other only indirectly (e.g., T1 and T3 via the sentence and section tokens). Right: Sparse attention matrix.

contextualization is performed iteratively over several layers. While hierarchical processing improves efficiency, separation of contextualization levels into different layers results in two problems: (1) Contextualization is not conducted simultaneously across all levels, hence more layers are required potentially. (2) Because different layers assume different functions, they can't serve as drop-in replacements for common Transformer architectures. HDT resolves these problems by simultaneous contextualization over all hierarchy levels in a *single layer* using an efficient hierarchical attention pattern.

Due to the quadratic memory and time complexity of the self-attention block of Transformer models (Vaswani et al., 2017), major efforts have been devoted to reducing complexity, in particular when applying attention to longer sequences. One branch of work focuses on using sparse attention patterns to reduce computation while maintaining expressiveness including Sparse Transformer (Rewon et al., 2019), ETC (Ainslie et al., 2020), BigBird (Zaheer et al., 2020), Longformer (Beltagy et al., 2020) and CoLT5 (Ainslie et al., 2023) among others. Another branch of work aims at speeding up attention computation by considering GPU hardware characteristics. FlashAttention 1 & 2 (Dao et al., 2022; Dao, 2023) propose tiling with block-wise attention to reduce memory IOs and compute attention blocks in parallel in Static Random Access Memory (SRAM). Matteo et al. (2023) extends the idea of FlashAttention to key/query dropping and hashing-based attention that was originally proposed by Kitaev et al. (2020). Our work builds upon both branches: We propose a dynamic sample-dependent sparse attention pattern which exploits the structure of text documents. For efficiency, we follow Dao et al. (2022) and implement this pattern as a customized memory-aware kernel.

## 3    Methodology

We now introduce the proposed Hierarchical Document Transformer (HDT) for efficient long-document modeling. First, we briefly recap the standard transformer model. Next, we introduce the proposed HDT block. Finally, we describe the design of the encoder-only as well as encoder-decoder HDT architectures which are used in our experiments in Section 4.

### 3.1    Standard Transformer

The original transformer encoder model by Vaswani et al. (2017) is composed of Multi-Head Self-Attention (MHSA) layers that are interleaved with shallow feed-forward networks and residual connections. Because the attention pattern is identical across multiple heads in an attention layer, for simplicity of notation, we represent queries, keys and values for a single attention head as $\mathbf{Q}, \mathbf{K}, \mathbf{V} \in \mathbb{R}^{n \times d_k}$, where $n$ is the sequence length and $d_k$ the head dimension. All MHSA layers in the model, as well as the embedding layers, produce outputs of dimension $d_{\text{model}}$. We further denote the number of heads as $h$, hence $d_{\text{model}} = d_k \times h$. A

| Tree Level | | | [DOC] | [SEC] | [SENT] | T1 | [SEC] | [SENT] | T2 | [SENT] | T3 | T4 | T5 | T6 |
|---|---|---|---|---|---|---|---|---|---|---|---|---|---|---|
| 1 - Section | $p^1$ | = | 0 | 1 | 1 | 1 | 2 | 2 | 2 | 2 | 2 | 2 | 2 | 2 |
| 2 - Sentence | $p^2$ | = | 0 | 0 | 1 | 1 | 0 | 1 | 1 | 2 | 2 | 2 | 2 | 2 |
| 3 - Token | $p^3$ | = | 0 | 0 | 0 | 1 | 0 | 0 | 1 | 0 | 1 | 2 | 3 | 4 |

**Figure 3: Hierarchical Positional Encoding.** We represent the position of each token in the hierarchy with one linear index $p^l$ per hierarchy level $l$ yielding an index vector $\mathbf{p} = (p^1, \ldots, p^L)^T$. Above, we show an example with $L = 3$ levels. Note that level 0 (document) does not require an index. Each index in $\mathbf{p}$ is passed through sinusoidal encoding functions which are summed over all levels to form the final encoding vector according to Eq. (2).

self-attention head computes its output $\mathbf{O} \in \mathbb{R}^{n \times d_k}$ as follows:

$$\mathbf{A} = \frac{\mathbf{QK}^T}{\sqrt{d_k}} \qquad \mathbf{S} = \text{softmax}(\mathbf{A}) \qquad \mathbf{O} = \mathbf{SV} \tag{1}$$

where the softmax operator is applied row-wise. The standard attention block has $O(n^2)$ time and memory complexity where $n$ is the input length. It further does not explicitly take the structure of documents into account.

## 3.2 Hierarchical Document Transformer

Most documents are organized into structural constituents like sections, paragraphs, sentences, bulleted lists, figures, and footnotes. This structure is represented in the visual layout and conveys the author's semantic organization of the text (Taylor & Beach, 1984; Guthrie et al., 1991). While our model is general and can handle arbitrary hierarchical structures, for simplicity we will focus our exposition on a document hierarchy with three levels: tokens, sentences and sections. More specifically, we split a document with $n$ tokens $\mathcal{D} = (t_1, t_2, ..., t_n)$ into sections $\mathcal{D} = (\mathcal{E}_1, \mathcal{E}_2, ..., \mathcal{E}_{|\mathcal{D}|})$. Sections $\mathcal{E}_i$ are split into sentences $\mathcal{E}_i = (\mathcal{S}_1, \mathcal{S}_2, ..., \mathcal{S}_{|\mathcal{E}_i|})$ which are split into sequences of regular tokens $\mathcal{S}_j = (t_1, t_2, ..., t_{|\mathcal{S}_j|})$.

We exploit this document structure by (1) introducing auxiliary anchor tokens to represent each element in the hierarchy, and (2) developing an efficient sparse attention kernel that exchanges information only between tokens at the same level (siblings) as well as between the respective parent and child tokens. By stacking multiple HDT blocks, information from any token can reach any other token. At the same time, the attention pattern of HDT blocks is highly sparse, leading to gains in computational (and memory) efficiency. More specifically, for the document hierarchy introduced above, we prepend additional [SENT] anchor tokens to the beginning of every sentence, [SEC] anchor tokens to the start of each section, and a [DOC] anchor token to the beginning of the document as illustrated in Fig. 2.

**Hierarchical Positional Encoding:** We extend the sinusoidal position encoding to model $L$ hierarchy levels. To inform each token about its position within the hierarchy, we assign it one linear index $p^l$ per hierarchy level as illustrated in Fig. 3, yielding an index vector $\mathbf{p} = (p^1, \ldots, p^L)^T$. Each index in $\mathbf{p}$ is passed through a set of standard sinusoidal encoding functions which are summed over all levels to form the final hierarchical positional encoding (HPE) vector at $\mathbf{p}$:

$$HPE(\mathbf{p}, i) = \sum_{l=1}^{L} \begin{cases} \sin(\omega_k p^l) & \text{if } i = 2k \\ \cos(\omega_k p^l) & \text{if } i = 2k+1 \end{cases} \quad \text{where} \quad \omega_k = \frac{1}{10000^{2k/d_{\text{model}}}} \tag{2}$$

**Hierarchical Attention:** To model document structure, we impose a document-specific attention pattern by modifying the standard Transformer attention in Eq. (1) as follows

$$\mathbf{A} = \frac{\mathbf{QK}^T}{\sqrt{d_k}} \qquad \mathbf{S} = \text{softmax}(\mathbf{A} \odot \mathbb{1}_\mathbf{M}) \qquad \mathbf{O} = \mathbf{SV} \tag{3}$$

with $(A_{ij} \odot \mathbb{1}_{M_{ij}}) = A_{ij}$ if $M_{ij} = 1$ and $-\infty$ if $M_{ij} = 0$. For clarity of notation, we use $p_i^l, p_j^l$ to represent the position index for hierarchy level $l$ of the $i$'th token $t_i$ and the $j$'th token

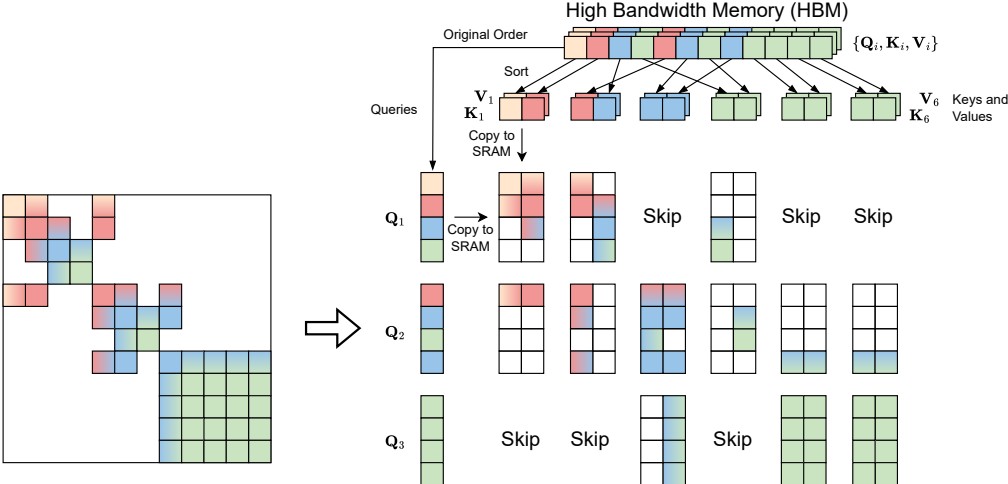

**Figure 4: Hierarchial Attention Kernel.** We copy queries, keys and values block-wise to SRAM for fast attention computation using a fused kernel. To increase the number of empty blocks that can be skipped, we sort keys and values based on their hierarchy level. Larger examples are shown in Fig. 10.

$t_j$, respectively. The attention mask $\mathbf{M} \in \{0, 1\}^{n \times n}$ is defined such that information is only exchanged directly between tokens at the same level as well as between the respective parent and child tokens. For a 3-level document structure, we first compute the attention mask for each hierarchy level separately

$$M_{ij}^{\text{DOC}} = [p_i^2 = 0] \cdot [p_j^2 = 0] \tag{4}$$

$$M_{ij}^{\text{SEC}} = [p_i^3 = 0] \cdot [p_j^3 = 0] \cdot [p_i^1 = p_j^1] \tag{5}$$

$$M_{ij}^{\text{SENT}} = [p_i^1 = p_j^1] \cdot [p_i^2 = p_j^2] \tag{6}$$

where $[\cdot]$ denotes the Iverson bracket which evaluates to 1 if the argument is true and 0 otherwise. Finally, we perform an OR operation to obtain the full attention mask $\mathbf{M}$:

$$\mathbf{M} = \mathbf{M}^{\text{DOC}} \oplus \mathbf{M}^{\text{SEC}} \oplus \mathbf{M}^{\text{SENT}} \tag{7}$$

Note that $\mathbf{M}$ is highly sparse in practice (see Fig. 10 for an example) and hence reduces theoretical complexity from $O(n^2)$ to $O(n\,s)$ where $s$ is the length of the longest sentence in the document. Thus, computational savings are largest for long documents for which $s \ll n$. However, when using parallel hardware (e.g., GPUs) as customary in deep learning, special care has to be taken regarding the kernel design to translate these theoretical savings into actual wall-clock time reduction. We hence develop a custom hierarchical attention kernel.

More specifically, we build upon the recent tiling-based ideas of FlashAttention (Dao et al., 2022; Dao, 2023). FlashAttention partitions attention computation into small $128 \times 64$ token *blocks* that can be computed efficiently in SRAM. This is in contrast to classical attention implementations that materialize intermediate outputs to slow High Bandwidth Memory (HBM). However, naïvely partitioning the attention matrix into regular blocks is suboptimal given the uniformly distributed entries of $\mathbf{M}$. Furthermore, each document has a different structure and hence leads to a different sparsity pattern in the attention matrix.

To maximize the number of empty blocks that can be skipped, we leverage a simple heuristic which is illustrated with an example in Fig. 4. Specifically, before copying keys and values to SRAM, we first sort them based on their hierarchy level from $l = 0$ to $L$ while keeping the order of the queries unchanged. This ensures adjacency of the most related tokens and hence increases the probability of large empty blocks that can be skipped as illustrated in Appendix Fig. 10. Afterwards, we copy the queries $\mathbf{Q}_i$, keys $\mathbf{K}_j$ and values $\mathbf{V}_j$ of block $(i, j)$ to SRAM and apply block attention. We process all non-empty blocks in parallel, skipping empty ones. Finally, we write the result $\mathbf{O}_i$ back to HBM using the online softmax

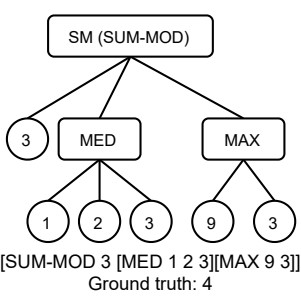

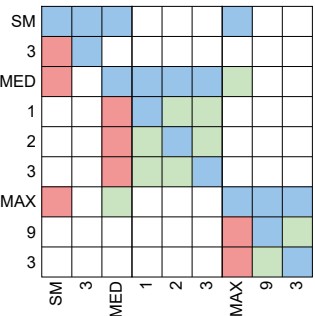

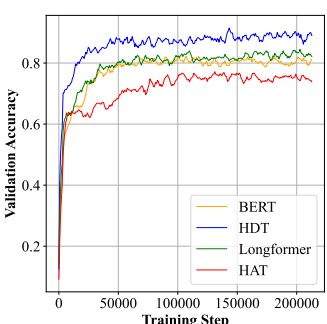

**Figure 5:** ListOps Sample  **Figure 6:** ListOps Attention  **Figure 7:** ListOps Convergence

algorithm (Milakov & Gimelshein, 2018). The complete algorithm for a forward pass is provided in Appendix Algorithm 1. While we found this simple sorting heuristic to work well in practice (see Fig. 8 for a runtime comparison to Block-Sparse FlashAttention and Appendix Fig. 10 for qualitative examples), more advanced permutation algorithms for sparse matrices (Aykanat et al., 2004; Ferris & Horn, 1998; Hendrickson & Kolda, 2000) could possibly lead to larger computational savings at the cost of additional inference time per data sample. We will leave the investigation of such trade-offs for future work.

**Hierarchical Encoder and Decoder Stacks:** We realize the *encoder* as a stack of $N$ identical HDT blocks where each block is composed of two sub-layers as illustrated in Fig. 1a. The first sub-layer performs hierarchical multi-head self-attention as described above, and the second is a simple position-wise fully connected feed-forward network (FFN). We use layer normalization at the beginning of each sub-layer as well as residual connections (He et al., 2016) for each layer. We apply the hierarchical positional encoding to the input tokens and add them to the respective token embeddings as input to the encoder stacks. The *decoder* is composed of a stack of $N$ identical standard Transformer decoder blocks, including a sub-layer of causal multi-head attention that prevents tokens from attending to subsequent positions, a sub-layer to perform multi-head cross attention wrt. the output of the encoder stack, and a standard feed-forward network (FFN). Following T5 (Raffel et al., 2020), we use relative positional encoding for the decoder.

## 4 Experiments

We first demonstrate the utility of structure-aware attention patterns on a simple mathematical task. Next, we show the effectiveness of our encoder-only model on SciRepEval proximity tasks. We also investigate the expressiveness of the anchor token representations using our encoder-decoder model on the FacetSum summarization tasks. Experiments on SCROLLS demonstrate that our model can even be applied to long texts which are not explicitly structured. Finally, we provide a detailed efficiency analysis.

### 4.1 Mathematical Reasoning Tasks

As proof of concept, we first compare our encoder model (HDT-E) to BERT (Devlin et al., 2019), Longformer (Beltagy et al., 2020) and HAT (Chalkidis et al., 2022) on the ListOps mathematical reasoning task (Nikita & R., 2018). The ListOps dataset is composed of simple list operations (Modular Sum, Minimium, Maximum, Median) with 2-5 operands and maximum tree depth 20. An example is shown in Fig. 5. Fig. 6 shows the different types of attention patterns we use. The combination of red+green+blue entries corresponds to the attention pattern defined in Section 3.2. However, the special structure of the problem admits of further increasing sparsity by using a "causal"

| Model | Acc. |
|---|---|
| BERT | 75.9 |
| HDT$_{r+g+b}$ | 79.7 |
| HDT$_{g+b}$ | 85.6 |
| HDT$_{blue}$ | **86.2** |

**Table 1:** ListOps Acc.

mask where operands do *not* attend to their operators (green+blue) and where operands do *not* attend to each other (blue only). Our ablations in Table 1 show that as we increase

the inductive sparsity bias, test set accuracy improves. As illustrated in Fig. 7, whilst BERT, Longformer and HAT use a learned positional embedding, we find that HDT outperforms all baselines on ListOps even without using any positional embedding as the operators in the ListOps task are invariant to the positions of their operands.

## 4.2 Language Tasks

We now investigate the utility of HDT for long-document processing tasks. As we are interested in efficiency, we conduct all experiments using a constrained compute budget. Following Geiping & Goldstein (2023), we pre-train our encoder models (HDT-E) for 24 hours on 1 GPU (∼10k steps) and our encoder-decoder models (HDT-ED) for 72 hours on 4 GPUs (∼50k steps) using a batch size of 128. Compared to the baselines, our models require 5-20 times fewer pre-training steps to reach better or comparable downstream task performance. We leave scaling HDT to larger sizes and more data for future work.

**Data:** To train our proposed structure-aware models, we build a large full-text document corpus from unarXive (Saier et al., 2023) (1.9M arXiv papers), HUPD (Suzgun et al., 2023) (utility patent applications) and the latest Wikipedia dump processed with Gensim (Řehůřek & Sojka, 2010). In total, our corpus includes over 12M long documents with extracted structural information. Our word tokenizer utilizes a vocabulary size of 32,768 tokens, trained with Byte Pair Encoding (BPE) (Sennrich et al., 2016). For splitting sentences, we use the NLTK sentence tokenizer[1]. We report results of HDT-E pre-trained with and without arXiv data on a full-document version of SciRepEval (Singh et al., 2023). We evaluate HDT-ED on the FacetSum (Meng et al., 2021) and SCROLLS (Shaham et al., 2022) benchmarks which contain tasks for long text reasoning requiring context modeling. Benchmarks details are provided in Appendix Section A.2.2.

**Baselines:** We compare HDT-E to Hierarchical Attention Transformer (HAT) (Chalkidis et al., 2022) which is the closest work to us that applies attention hierarchically to reduce computation/memory usage for processing long text. HDT-E differs from HAT as follows: (1) HDT-E takes the natural structure of text to build hierarchies while HAT cuts long text into fixed-size segmentations that do not adheare to the structure of natural language, (2) HDT-E enables simultaneous information passing across all levels in a single layer while HAT requires multiple blocks to contextualize information between hierarchies. Moreover, HAT only considers two hierarchy levels. Despite the flexibility of our model, it achieves comparable latency/throughput supported due to our highly efficient kernel implementation. As the sparse attention baseline, we choose Longformer (Beltagy et al., 2020) and its encoder-decoder variant Longformer-Encoder-Decoder (LED) which are effective on various long-text tasks in previous works (Tay et al., 2021; Dasigi et al., 2021; Tay et al., 2023a). We also include the current SotA methods SciBERT (Beltagy et al., 2019) and SciNCL (Ostendorff et al., 2022) as additional SciRepEval baselines. Both baselines use dense attention with an input length of 512 capturing only title and abstract. Unless stated otherwise, we use the officially released code and models by the original authors.

**Pre-training & Fine-tuning:** We pre-train the encoder-only model HDT-E and the encoder-decoder model HDT-ED from scratch on our training corpus. All models are pre-trained using an input length of 8,192 tokens and a mini-batch size of 128 (via gradient accumulation). HDT-E is pre-trained on the standard Masked Language Modeling (MLM) objective with a mask ratio of 15%. We also pre-train Longformer from scratch in the same setting as ours to study the effectiveness of the attention pattern we propose. Fig. 1b shows that our model converges faster than Longformer. HDT-ED is pre-trained on UL2 (Tay et al., 2023b), which is a unified pre-training paradigm with a range of denoising tasks. We fine-tune all models on the downstream tasks using the default settings, see Section A.2.2 for details.

**Results on SciRepEval (Encoder Models):** SciRepEval (Singh et al., 2023) is a scientific document representation benchmark containing various classification, regression, and proximity tasks. Unfortunately, the original SciRepEval dataset comprises only titles and abstracts as for many papers in the original benchmark the full text is not publicly available.

---

[1]https://www.nltk.org/api/nltk.tokenize.html

| Model | Full Text | \multicolumn{4}{c}{SciDocs} | | | | Feeds-M | High. Infl. | Avg. |
| | | Cite | CoCite | CoView | CoRead | | | |
| --- | --- | --- | --- | --- | --- | --- | --- | --- |
| **Pretrained Only** | | | | | | | | |
| SciBERT_base | | 53.75 | 66.73 | 66.37 | 53.20 | 63.18 | 40.80 | 57.34 |
| Longformer | ✓ | 56.64 | 71.92 | 71.51 | 61.57 | 63.66 | 43.82 | 61.52 |
| HAT | ✓ | 60.14 | 75.55 | 73.62 | 67.65 | 65.02 | **45.81** | 64.63 |
| HDT-E | ✓ | **62.50** | **78.51** | **75.69** | **72.12** | 65.19 | 43.60 | **66.27** |
| HDT-E (-arXiv) | ✓ | 59.03 | 76.05 | 72.85 | 71.71 | **65.41** | 43.69 | 64.79 |
| **Pretrained + Finetuned with Contrastive Learning** | | | | | | | | |
| SciNCL @684k | | 64.77 | 81.67 | 78.55 | 77.48 | 70.22 | 48.66 | 70.23 |
| SciNCL @19k | | 62.56 | 82.29 | 77.84 | 75.84 | 67.11 | 46.23 | 68.65 |
| Longformer @19k | ✓ | 61.75 | 79.87 | 78.20 | 74.25 | 67.80 | 43.85 | 67.62 |
| HAT @19k | ✓ | 63.46 | 81.24 | **79.43** | 75.76 | 69.31 | 47.37 | 69.42 |
| HDT-E @19k | ✓ | **64.23** | **82.44** | 78.95 | **77.09** | **71.22** | **49.37** | **70.55** |
| HDT-E @19k (-arXiv) | ✓ | 63.34 | 82.18 | 79.06 | 76.78 | 70.64 | 48.95 | 70.16 |

**Table 2: Results on SciRepEval Proximity Tasks.** Top: Models pre-trained with MLM without fine-tuning. Bottom: Models pre-trained with MLM and fine-tuned using SciNCL's contrastive learning objective. Full text documents are available only for a subset of 19k training triplets. For reference, we also report the results of the original SciNCL model which is trained on all 684k title+abstract triplets. We also report HDT-E pre-trained without arXiv data (-arXiv) to study the impact of scientific documents as pre-training data to our model's performance on the SciRepEval tasks which are in the scientific domain. All numbers are mean average precision. SciBERT and SciNCL use only title and abstract as input.

To adapt SciRepEval for long document models, we hence consider a subset of SciRepEval for which full-text articles are available in unarXive (Saier et al., 2023). Table 2 shows the tasks with the largest number of samples, which are proximity tasks involving ranking a set of candidate papers by their relatedness to a query paper. Results on other SciRepEval tasks can be found in Appendix Table 6. Without any fine-tuning, we take the hidden representation of the first token (either [CLS] for baseline models or [DOC] for HDT-E) and sort papers by cosine similarity between paper representations. The first set of results in Table 2 ("Pretrained Only") shows the mean average precision for models pre-trained on MLM. Our results demonstrate that both, pre-training on full-text documents and considering the structure of documents, leads to substantial performance gains in this setting. Moreover, our model pre-trains significantly faster (10k iterations) than Longformer (64k) and HAT (50k) and can be trained from scratch while Longformer and HAT require parameter initialization from BART (Lewis et al., 2020) and RoBERTa (Liu et al., 2019) (respectively) for best performance. This result underscores the sample efficiency of our structure-aware HDT model.

Currently, the state-of-the-art performance on SciRepEval is reported by SciNCL (Ostendorff et al., 2022) which finetunes a pre-trained SciBERT (Beltagy et al., 2019) model using a contrastive learning objective. When comparing in this setting ("Pretrained + Finetuned with Contrastive Learning"), we observe that HDT-E again outperforms all baselines, despite with a smaller margin. Notably, HDT-E outperforms SciNCL even though SciNCL has access to 684k triplets for finetuning while HDT-E is trained only on those 19k triplets for which full-text documents are publicly available. In addition, to test our model's reliance on scientific documents for pre-training, we *exclude* the entire arXiv corpus from our pre-training dataset. We observed that our model's performance nearly matches HAT in the pre-training-only setting and surpasses HAT in the contrastive learning setting, even when using only out-of-domain data for pre-training.

**Results on FacetSum (Encoder-Decoder Models):** A major advantage of our model is its flexibility. As tokens are hierarchically grouped via anchor tokens, the document's content is compressed at different levels of granularity. We can query this information by letting the decoder of HDT-ED attends to only a subset of the tokens or anchor tokens (e.g., [SEC]). To better understand how much information is retained where in our model, we design a

| Model | Purpose | Method | Findings | Value | Purpose-ZS | Method-ZS |
|---|---|---|---|---|---|---|
| LED$_{\text{base}}$ | 39.51 | 19.31 | 19.22 | **24.26** | 18.54 | 14.12 |
| HDT-ED-[SEC] | 30.70 | 17.65 | 17.15 | 18.90 | 15.01 | 12.67 |
| +[SENT] | 34.29 | 19.72 | 18.38 | 20.53 | 19.67 | 13.94 |
| + tokens | **40.60** | **22.21** | **22.11** | 22.11 | **21.75** | **15.43** |

**Table 3: Results on FacetSum Summarization Task.** Following the original paper, we report ROUGE-L as the metric here. For HDT-ED-[SEC], the decoder cross-attends only to the section anchor token [SEC]. We observe that even when attending only to the anchor tokens (+[SENT]), our model is on par with LED, where the decoder attends to all tokens of the section, demonstrating the expressiveness of the learned intermediate representation of anchor tokens. When attending to additional regular tokens (+tokens), our model outperforms LED. We also report zero-shot (ZS) performance for "Purpose" and "Method", training only on "Findings" and "Value".

| Model | GovRep | SumScr | QMSum | Qspr | Nrtv | QALT | CNLI | Avg |
|---|---|---|---|---|---|---|---|---|
| | ROUGE-1/2/L | ROUGE-1/2/L | ROUGE-1/2/L | F1 | F1 | EM-T/H | EM | Score |
| LED$_{\text{base}}$ | **56.2/26.6/28.8** | 24.2/4.5/15.4 | 25.1/**6.7/18.8** | 26.6 | **18.5** | 25.8/25.4 | 71.5 | 29.16 |
| HDT-ED | 49.8/22.2/25.8 | **30.8/7.1/18.6** | **28.3/6.7**/18.7 | **33.1** | 14.2 | **29.4/26.4** | **81.4** | **31.41** |

**Table 4: Results on the SCROLLS Summarization, QA and NLI Benchmark.** We compare HDT-ED (pre-trained for 12 GPU days) to Longformer-Encoder-Decoder (LED) on the official SCROLLS benchmark *without document structure*. We choose LED as baseline as it has a comparable number of parameters (162M) to HDT-ED (124M). We remark that neither model is competitive with state-of-the-art billion-parameter models such as CoLT5 XL (score 43.5) which are trained on large GPU clusters.

summarization task based on FacetSum (Meng et al., 2021) which provides section-level summaries for Emerald journal articles. Table 3 shows our results. We observe performance on par with LED$_{\text{base}}$ even when attending only to the anchor tokens, which demonstrates the expressiveness of the learned intermediate representations. When additionally attending to all regular tokens (as LED$_{\text{base}}$ does), performance increases further. The right part of the table shows generalization performance when training on "Findings" and "Value", but evaluating "Purpose" and "Method" summaries.

**Results on SCROLLS (Encoder-Decoder Models):** While our model benefits from the structure of documents as present in our SciRepEval and FacetSum evaluations, we are also interested in the applicability of our model to less structured "flat" long-text tasks. Towards this goal, we evaluate our model on the SCROLLS benchmark, which contains 7 tasks spanning multiple domains that require reasoning over long texts for which document structure is not available. To apply our model to such inputs, we use pseudo sections composed of a fixed number of 32 sentences for hierarchical attention, while sentences are extracted as usual. We fine-tune HDT-ED on the 7 tasks in SCROLLS separately and submit the test set predictions to the public leaderboard[2]. Table 4 show our results. Compared to LED$_{\text{base}}$ with a similar number of parameters, HDT-ED improves by over 2 score points. This demonstrates that our model is also effective on "flat" long-text tasks. However, unsurprisingly, our results are not competitive with state-of-the-art billion-parameter models such as CoLT5 XL which are trained for many epochs on large industrial GPU clusters.

**Efficiency Analysis:** Denoting the length (number of tokens) of the longest sentence in a document as $s$, the theoretical complexity of HDT attention is $O(n \times s)$. Fig. 8 compares GPU runtime and memory usage of different attention layers, including standard dense attention, block-sparse FlashAttention (using our pattern), Longformer sparse windowed attention, and HDT attention. All kernels are fed with real data, i.e., documents, which are further transformed to multi-head queries, keys, and values using 12 heads with head dimension 64. We report the runtime for one batch with 4 samples on an NVIDIA A100 GPU, and plot peak GPU memory consumption at lengths 16k and 32k for each kernel, except standard attention which exceeds 40 GB at 16k tokens. Besides, we also compare runtime and memory consumption for three complete 12-layer models using different attention mechanisms in Table 5. Despite its flexibility, HDT-E achieves runtime and memory consumption on par with HAT which uses 2 layers and fixed segment length.

---

[2]https://www.scrolls-benchmark.com/leaderboard

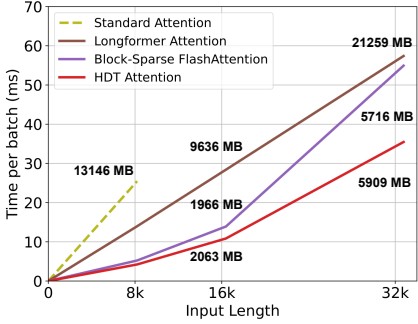

**Figure 8: Runtime and Memory** consumption of a *single attention layer*.

| Model | Longformer | HAT | HDT-E |
|---|---|---|---|
| **Complex.** | $O(n\,w)$ | $O(n\,k)$ | $O(n\,s)$ |
| **#Params** | 148.66 M | 152.73 M | **108.99 M** |
| **Time (ms)** | $178.82 \pm 6.84$ | **77.84** $\pm 2.30$ | $79.8 \pm 2.96$ |
| **TFLOPS** | $5.29 \pm 0.19$ | $8.95 \pm 0.26$ | **8.99** $\pm 0.34$ |
| **Memory** | 11.25 GB | **5.3 GB** | 5.85 GB |

**Table 5: Runtime and Memory** consumption of several *long-document models* with 12 layers. We report complexity, parameters, inference time, throughput, and memory usage using context length $n = 4096$ and mini-batch size 1. Here, $w = 512$ is the Longformer window size, $k = 128$ is the fixed HAT segment length. $s$ is the length of the longest sentence in the document.

## 5 Conclusion

We presented Hierarchical Document Transformer (HDT), a novel approach for encoding long documents efficiently. By explicitly incorporating document structure into the attention mechanism, we achieve sparse representations, reducing computational complexity while improving sample efficiency and generalization. We believe that hierarchical text representations offer many exciting opportunities in the future: Extending the hierarchical structure down to byte-level could enable token-free language models. Hierarchical ideas might also inspire novel decoder architectures that generate language in a structured hierarchical fashion. Hierarchical models may also be advantageously combined with other models such as state space models, RNNs or ConvNets. Finally, it still remains unclear if scaling laws also hold for hierarchical language models as they do for LLMs, and if hierarchical language models hold similar potential for unlocking emergent abilities.

**Acknowledgments**

Andreas Geiger is a member of the Machine Learning Cluster of Excellence, funded by the Deutsche Forschungsgemeinschaft (DFG, German Research Foundation) under Germany's Excellence Strategy – EXC number 2064/1 – Project number 390727645. Jan Buchmann was funded by the European Union (ERC, InterText, 101054961). Views and opinions expressed are however those of the author(s) only and do not necessarily reflect those of the European Union or the European Research Council. Neither the European Union nor the granting authority can be held responsible for them.

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

## A    Appendix

### A.1    Hierarchical Attention Kernel

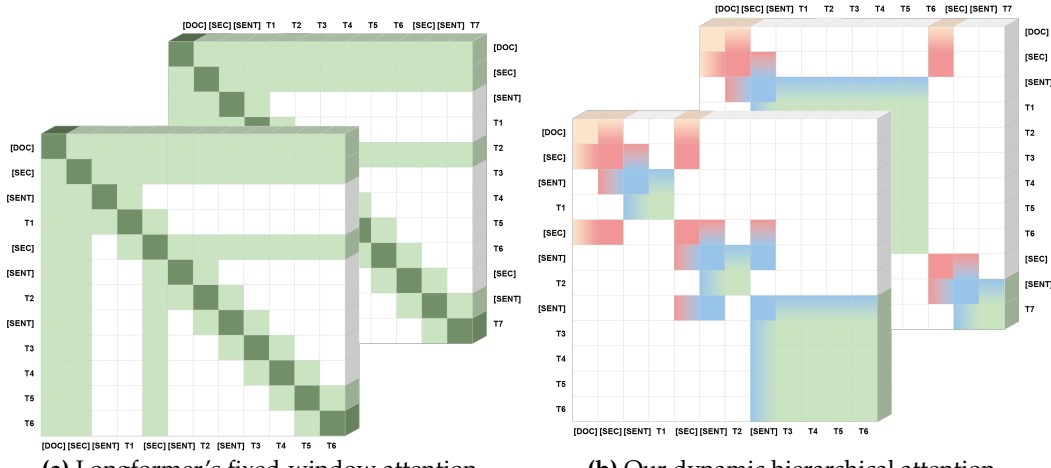

**(a)** Longformer's fixed-window attention    **(b)** Our dynamic hierarchical attention

**Figure 9:** (a) The Longformer (Beltagy et al., 2020) sparse attention pattern is identical for all samples in a mini-batch. (b) In contrast, the proposed dynamic hierarchical attention pattern considers the document structure and hence is different for each sample in a mini-batch.

In this section, we compare the proposed dynamic hierarchical attention pattern to the fixed sparse attention pattern of Longformer and provide algorithmic details of its implementation. Subsequently, we illustrate how our customized attention kernel reduces computational overhead through qualitative examples.

Given that our attention pattern is rooted in document structure, it is different for each sample. Hence, we require an attention mechanism that is able to cope with sample-dependent attention patterns. Fig. 9 provides a visual illustration comparing our dynamic attention pattern to the fixed attention pattern of Longformer (Beltagy et al., 2020), assuming a mini-batch size of 2 for clarity. Notably, within a mini-batch containing multiple samples, the attention pattern applied to each sample varies, thereby complicating the realization of our attention pattern using existing attention kernels. Moreover, due to the presence of anchor tokens positioned at the onset of different hierarchies and attending to one another, the attention devoted to these anchor tokens can be notably fine-grained and sparse within

the attention mask. This phenomenon is demonstrated in Fig. 10, depicting hierarchical attention masks derived from real documents. Note that due to space limitations only the first 1k tokens (∼25% of the total document size) are visualized. To optimize the number of empty blocks that can be skipped, we employ the sorting heuristic outlined in Section 3.2. The core mechanism of our hierarchical sparse attention kernel is based on and extends FlashAttention Dao et al. (2022). Our algorithm for a forward pass of the attention layer is provided in Algorithm 1. This algorithm is executed in parallel for all elements of the Cartesian product $B \times H$, where $B$ represents the batch size and $H$ denotes the number of heads, assuming input tensors of shape $(B \times H \times n \times d_k)$.

Our primary adjustment to the FlashAttention algorithm is in line 2 of Algorithm1, where we implement a sorting mechanism for the key and value based on their hierarchy levels, specifically prioritizing [DOC] tokens followed by [SEC] and [SENT] tokens before normal tokens. This modification is grounded in the observation that the computational ineffi-ciency in block-wise hierarchical attention predominantly stems from attention interactions between anchor tokens situated distantly from each other, resulting in excessively sparse blocks. Fig. 10 offers a visual comparison between the computation patterns of our kernel and block-sparse FlashAttention, demonstrating the efficacy of our approach.

---

**Algorithm 1** Forward Pass of the HDT Hierarchical Attention Kernel

---

**Require:** Query, key, value $\mathbf{Q}, \mathbf{K}, \mathbf{V} \in \mathbb{R}^{n \times d_k}$, query block size $M$, key-value block size $N$, softmax statistics vectors $\mathbf{m} \in \mathbb{R}^{n \times n}$ and $\mathbf{l} \in \mathbb{R}^{n \times n}$, output tensor $\mathbf{O}^{n \times d_k}$.

1: Partition $\mathbf{Q}$ into $T_r = \lceil \frac{n}{M} \rceil$ blocks $\mathbf{Q}_1, \ldots, \mathbf{Q}_{T_r}$
2: Sort $\mathbf{K}, \mathbf{V}$ according to the hierarchy level from $l = 1$ to $L$.
3: Partition $\mathbf{K}, \mathbf{V}$ into $T_c = \lceil \frac{n}{N} \rceil$ blocks $\mathbf{K}_1, \ldots, \mathbf{K}_{T_c}$, and $\mathbf{V}_1, \ldots, \mathbf{V}_{T_c}$
4: **for** $1 \leq i \leq T_r$ **do**
5:     Load $\mathbf{Q}_i, \mathbf{O}_i, m_i, l_i$ from HBM to SRAM
6:     **for** $1 \leq j \leq T_c$ **do**
7:         Load $\mathbf{K}_j, \mathbf{V}_j$ from HBM to SRAM
8:         On chip, compute mask $\mathbf{M}$ according to Equation (7)
9:         **if** Non-zero values in $\mathbf{M}$ **then**
10:             $\mathbf{S}_{ij} = \mathbf{Q}_i \mathbf{K}_j \odot \mathbf{M}_{ij}$
11:             $\tilde{m}_{ij} = \text{rowmax}(\mathbf{S}_{ij})$
12:             $\tilde{\mathbf{P}}_{ij} = \exp(\mathbf{S}_{ij} - \tilde{m}_{ij})$ (pointwise)
13:             $\tilde{l}_{ij} = \text{rowsum}(\tilde{\mathbf{P}}_{ij})$
14:             $m_i^{\text{new}} = \max(m_i, \tilde{m}_{ij}) \in \mathbb{R}^M, l_i^{\text{new}} = e^{m_i - m_i^{\text{new}}} l_i + e^{\tilde{m}_{ij} - m_i^{\text{new}}} \tilde{l}_{i,j} \in \mathbb{R}^M$
15:             Write $\mathbf{O}_i \leftarrow \text{diag}(l_i^{new})^{-1}(\text{diag}(l_i) e^{m_i - m_i^{new}} \mathbf{O}_i + e^{\tilde{m}_{ij} - m_i^{new}} \tilde{\mathbf{P}}_{ij} \mathbf{V}_j)$ to HBM
16:             Write $l_i \leftarrow l_i^{new}, m_i \leftarrow m_i^{new}$ to HBM
17:         **end if**
18:     **end for**
19: **end for**

---

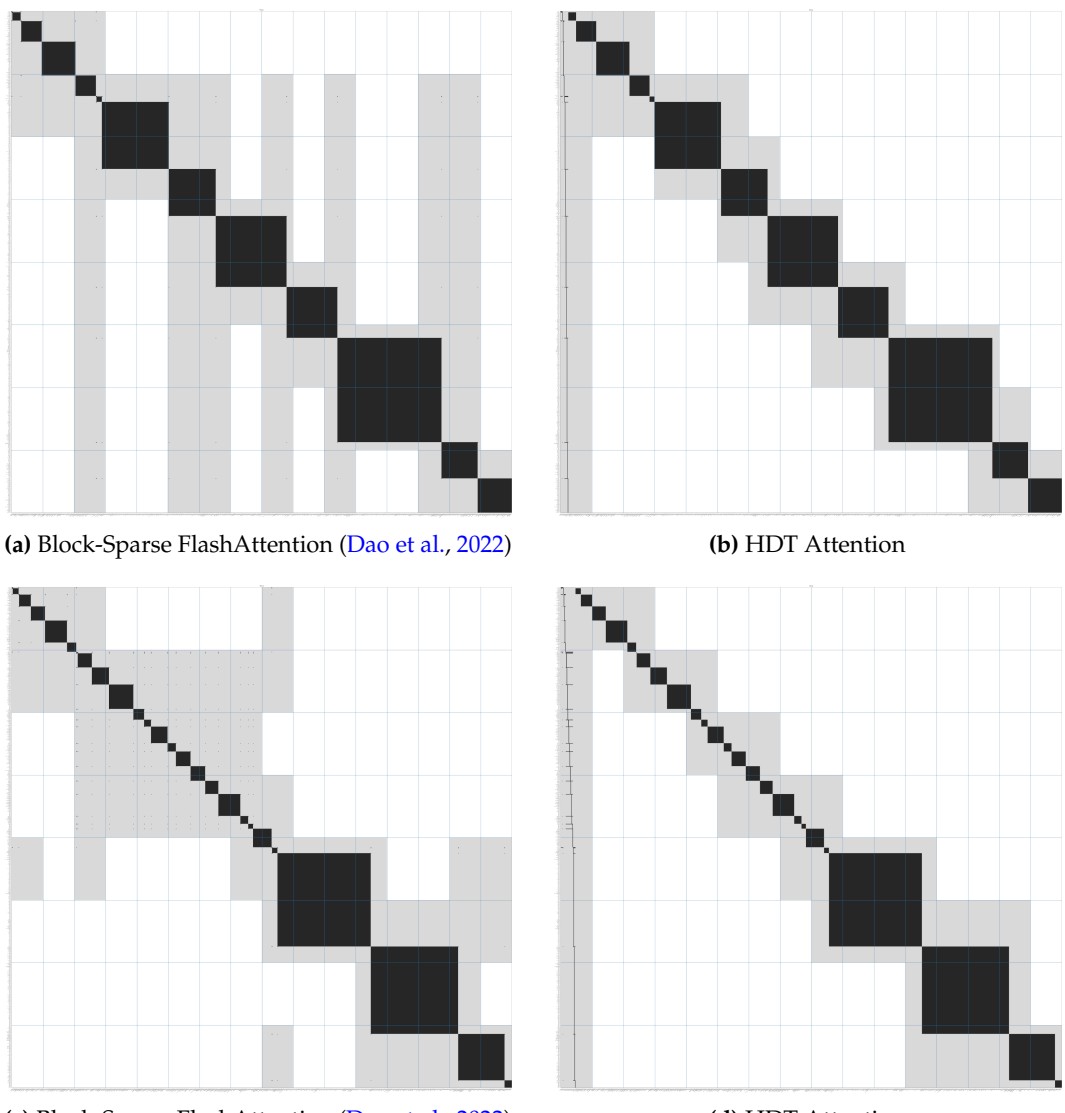

**(a)** Block-Sparse FlashAttention (Dao et al., 2022)    **(b)** HDT Attention

**(c)** Block-Sparse FlashAttention (Dao et al., 2022)    **(d)** HDT Attention

**Figure 10: Attention Pattern (Black) and Processed SRAM Blocks (Grey).** Comparison between the practical computation of Block-Sparse FlashAttention (Dao et al., 2022) and our HDT attention kernel on the same hierarchical attention patterns. We show the attention mask of the first 1k tokens (~25% of the total document size) of two different documents (row 1+2) in black. The blue grid illustrates the $128 \times 64$ SRAM blocks which are processed in parallel using the fused kernel. All blocks highlighted in grey contain at least one non-zero attention entry and hence require processing. Due to the reordering of keys and values (columns) in HDT, anchor tokens are aggregated within adjacent blocks leading to a larger number of blocks that can be skipped compared to Block-Sparse FlashAttention (Dao et al., 2022).

## A.2  Experiments

In this section, we provide additional details about the model settings and datasets.

### A.2.1  Mathematical Reasoning Tasks

**ListOps:** The models we train on ListOPs 20d have 12 transformer encoder layers with a feature dimension of 128, an intermediate size of 512, a learning rate of 3e-4 and a batch size of 200. We split the original training set into 85k samples for training and 5k samples for validation. The test set accuracy is computed on 10k samples. The ListOps code (Nikita & R., 2018) generates many short samples and a few very long ones, occasionally exceeding the 512 token input size of the BERT model. The largest token length that can be generated by ListOps 20d is $5^{20}$, with a maximum tree depth of 20 and the maximum number of operands per operator being 5. We exclude such rare samples from the dataset for our experiments. We observe that our conclusions regarding the benefits of sparse structure-aware attention patterns also hold on ListOps data generated with a smaller maximum tree depth of 10d and 5d. As shown in Fig. 11, HDT is capable of correctly predicting long ListOPs sequences, whereas BERT, Longformer and HAT exhibit lower performance for long samples. HDT's performance on ListOPs is stable across varying input lengths.

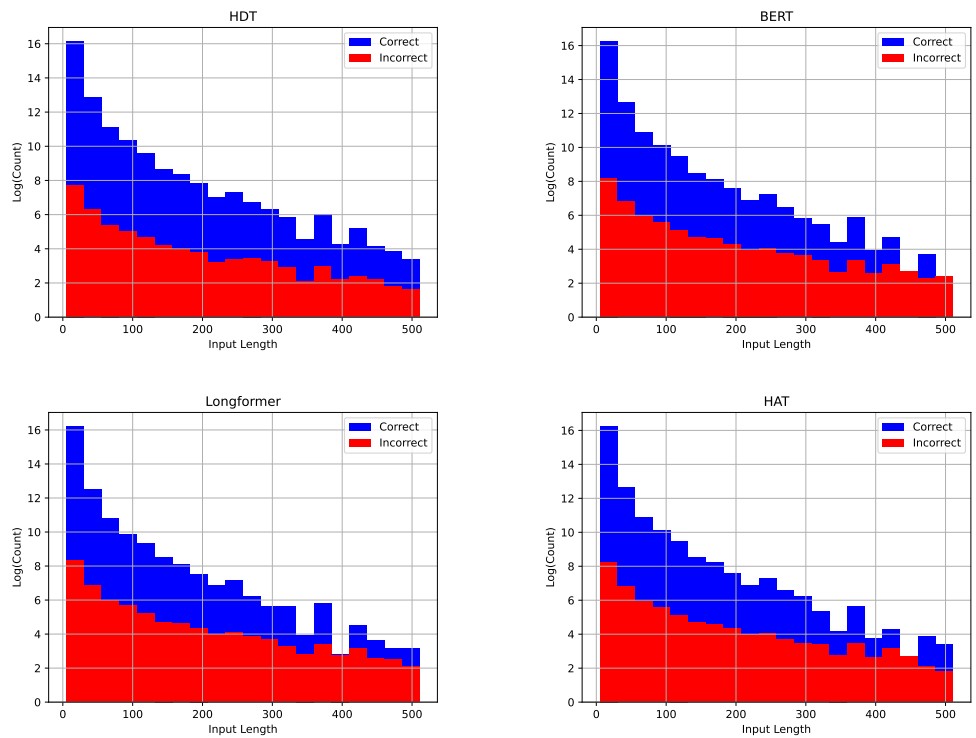

**Figure 11:** Stacked histograms of predictions by HDT, BERT, Longformer and HAT on ListOPs 20d.

### A.2.2  Language Tasks

**Model Settings:** We choose a typical setting for our model architecture, with $d_{\text{model}} = 768$, $d_k = 64$ with 12 heads, and an intermediate hidden size of 3,072. For HDT-E, we use 12 encoder stacks, and for HDT-ED we use 6 encoder stacks and 6 decoder stacks, resulting in 109M and 124M parameters, respectively. For fine-tuning, we use a constant learning rate of 5e-5, batch size 32 (by accumulating gradients of 8 mini-batches with mini-batch size 4), and a dropout rate of 0.1 for all tasks, except for summarization tasks for which we use a larger learning rate of 1e-3 which we empirically find better suited for all models in

| Model | SRH Search nDCG | PRX Feeds-1 MAP | Peer Review Score Kendall's $\mathcal{T}$ | Tweet Mentions Kendall's $\mathcal{T}$ | RGN hIndex Kendall's $\mathcal{T}$ | Citation Count Kendall's $\mathcal{T}$ | Publication Year Kendall's $\mathcal{T}$ | Average |
|---|---|---|---|---|---|---|---|---|
| SciBERT | 56.79 | 69.2 | **17.39** | 10.09 | **16.84** | 15.43 | 13.37 | 28.44 |
| Longformer | 56.11 | 66.46 | 17.34 | 12.68 | 16.65 | 26.16 | **25.43** | 31.54 |
| SciNCL | 56.05 | **71.85** | 13.15 | **13.95** | 11.92 | 26.27 | 18.69 | 30.27 |
| HAT | **57.01** | 70.01 | 14.62 | 13.18 | 13.64 | 19.75 | 19.8 | 29.72 |
| HDT-E (title+abstract) | 56.62 | 71.65 | 13.79 | 13.62 | 15.65 | **26.34** | 24.66 | **31.76** |

**Table 6: Results on remaining SciRepEval tasks.** Although our model is not the best across all tasks, the performance is comparably stable in different tasks than other models.

| Model | MIMIC F1 | ECtHR-LJP F1 | ECtHR-ARG acc |
|---|---|---|---|
| Longformer | 78.7 | 78.6 | 66.7 |
| HAT | 78.9 | **79.8** | 82.6 |
| HDT-E | **79.1** | 79.5 | **82.9** |

**Table 7: Results on document classification task ECtHR-LJP, ECtHR-ARG, MIMIC.** We follow the experimental setting of HAT and cover three long document classification tasks and report the test scores.

our experiments. We use AdamW (Loshchilov & Hutter, 2019) as the optimizer for both pre-training and fine-tuning.

**SciRepEval:** The SciRepEval benchmark we use in this work is a subset of the original dataset (Singh et al., 2023) with additional full-text data from unarXive (Saier et al., 2023). We remove samples that are not in unarXive and drop the tasks that have less than 100 samples left after filtering. This leads to a benchmark spanning 13 tasks, including regression, proximity, and search tasks for scientific document representation evaluation. We show our results on proximity tasks in Table 2 and results on the remaining tasks in Table 6.

**Long Document Classification Tasks:** Following HAT (Chalkidis et al., 2022), we validate our model on three additional long document classification tasks. The first task is MIMIC-III (Johnson et al., 2016), which comprises nearly 50,000 discharge summaries from US hospitals. Each summary is annotated with one or more ICD-9 codes (labels). The model takes a discharge summary as input and outputs the relevant set of first-level ICD-9 codes (19 in total). The second task is ECtHR-LJP (Chalkidis et al., 2021), containing approximately 11,000 cases from the European Court of Human Rights (ECtHR) public database. For each case, the dataset provides a list of factual paragraphs (facts) from the case description. Each case is mapped to articles of the ECHR that were allegedly violated. The model takes the list of facts as input and outputs the set of allegedly violated articles. Lastly, we include the ECtHR-ARG task (Habernal et al., 2023), which includes approximately 300 cases from the European Court of Human Rights (ECtHR). For each case, the dataset provides a list of argumentative paragraphs from the case analysis, with spans in each paragraph labeled with one or more of 13 argument types. We follow HAT in re-formulating this task as a sequential paragraph classification task, where each paragraph is labeled with one or more argument types. The model takes the list of paragraphs of a case as input and outputs the set of relevant argument types for each paragraph. Our results in Table 7 show that our model is comparable to HAT on this task.

**FacetSum:** The FacetSum dataset provides fixed 4-class summaries, "Purpose", "Method", "Findings", and "Value", for each document. To align classes to the corresponding sections, we follow Meng et al. (2021) to classify sections into "Introduction", "Method", "Result" and "Conclusion" first by keyword matching and then match the four classes of sections to the four classes of summaries, respectively.

**SCROLLS:** The SCROLLS benchmark contains a suite of tasks that require reasoning over long texts. The tasks cover GovReport (Huang et al., 2021) and SummScreenFD (Chen et al., 2022) that are summarization tasks in the domain of government reports and TV shows; QMSum (Zhong et al., 2021), a query-based summarization task for meeting transcripts; QASPER (Dasigi et al., 2021), a question answering dataset for scientific papers;

| Model | GovRep ROUGE-1/2/L | SumScr ROUGE-1/2/L | QMSum ROUGE-1/2/L | Qspr F1 | Nrtv F1 | QALT EM-T/H | CNLI EM | Avg |
|---|---|---|---|---|---|---|---|---|
| CoLT5$^{\dagger}_{\text{XL}}$ | 61.3/32.2/33.8 | 36.4/10.1/21.7 | 36.2/12.9/24.2 | 53.9 | 31.1 | 48.1/43.8 | 88.4 | 43.51 |
| LongT5$^{\dagger}_{\text{XL}}$ | 61.1/32.3/33.7 | 35.8/9.6/21.1 | 34.9/11.8/23.5 | 53.1 | 29.3 | 46.0/42.1 | 88.2 | 42.53 |
| CoLT5$^{\dagger}_{\text{Large}}$ | 60.7/31.3/32.9 | 36.7/10.6/22.0 | 34.9/11.5/23.1 | 49.8 | 27.7 | 39.9/36.8 | 88.7 | 41.04 |
| LED$^{\dagger}_{\text{base}}$ | 56.2/26.6/28.8 | 24.2/4.5/15.4 | 25.1/6.7/18.8 | 26.6 | 18.5 | 25.8/25.4 | 71.5 | 29.16 |
| HDT-ED | 49.8/22.2/25.8 | 30.8/7.1/18.6 | 28.3/6.7/18.7 | 33.1 | 14.2 | 29.4/26.4 | 81.4 | 31.41 |

**Table 8: Results on the SCROLLS benchmark compared to SotA.** We evaluate several models on the SCROLLS benchmark. † indicates results reported on the public leaderboard.

| Model | GovRep ROUGE-1/2/L | QASPER F1 | | | | |
|---|---|---|---|---|---|---|
| | | Extractive | Abstractive | Yes/No | Unanswer. | Overall |
| LED | 56.2/26.6/28.8 | 30.96 | 15.76 | 70.33 | 26.21 | 32.80 |
| HDT-ED | 49.89/21.54/25.26 | 30.57 | 11.42 | 67.14 | 46.45 | 33.14 |
| + struct. | 49.42/21.22/24.97 | 33.12 | 13.02 | 64.19 | 43.17 | 34.02 |

**Table 9: Effect of using document structure on GovRep and QASPER.** Note that the models are evaluated with the original GovRep (Huang et al., 2021) and QASPER (Dasigi et al., 2021) datasets, therefore results in this table might be slightly different from the results in Table 4 for the same task as the authors have cleaned the datasets in SCROLLS.

NarrativeQA (Kocisky et al., 2018), a question answering dataset over entire books from Project Gutenberg[3]; QuALITY (Pang et al., 2022), a multiple choice question answering dataset over stories and articles sourced from Project Gutenberg; and Contract NLI (Koreeda & Manning, 2021) as a natural language inference dataset in the legal domain. Table 8 compares our model with SotA billion-parameter encoder-decoder models which have been trained on large industrial compute clusters.

**Effect of document structure:** Our models are initially pre-trained on structured documents, and since most downstream tasks also maintain document structure, we aim to investigate the impact of such structures. To accomplish this, we selected two tasks, GovReport (Huang et al., 2021) and QASPER (Dasigi et al., 2021), where document structure preservation is integral. We proceeded by flattening the documents to create a flattened version of the dataset. Subsequently, we trained both LED and HDT-E on these datasets, employing the pseudo-section setting introduced in Section 4.2 for the flattened data. A comparative analysis between models trained with and without real document structure for the two datasets is presented in Table 9. While we observe that modeling document structure is not having a very large effect on downstream task performance in this setting, it is yet very important to utilize document structure during pre-training in our experiments. In fact, models pre-trained on pseudo sections and tested on the structured downstream tasks did not deliver any reasonable results. Our interpretation is that during pre-training, HDT learns to represent hierarchical information in different granularity via its anchor tokens, which allows it to adapt to pseudo-section data via fine-tuning.

---

[3]https://www.gutenberg.org/

