# OpenReview forum: "HDT: Hierarchical Document Transformer"
_colmweb.org/COLM/2024/Conference — COLM_

### Official Review · Reviewer_vjsH · 2024-05-09

**Rating:** 7
**Confidence:** 4
**Ethics Flag:** 1

**Summary:**

This paper describes a sparse Transformer architecture tailored for structured hierarchical documents (document->section->sentence->token). The new architecture introduces auxiliary anchor tokens (indicating the boundary of document/section/sentence) and redesigns the attention mechanism into a sparse multi-level hierarchy (ensuring each token only attending to its parent, siblings and children).
The authors point out that there are two main benefits of the hierarchical document transformer: exploiting the document structure as an inductive bias (better performance on downstream tasks); and, sparse attention, leading to gains in computational and memory efficiency (faster pre-training).
Additionally, the authors have created an efficient attention kernel using the Triton library, which can seamlessly replace existing Transformers. The implementation aims to minimize empty block computations by employing a simple heuristic: sorting keys and values according to their hierarchy level first.
To demonstrate the effectiveness of proposed structure-aware attention, the authors conduct a series of experiments on different datasets, including ListOps (mathematical reasoning), SciRepEval, FacetSum, SCROLLS. The authors show that their method outperforms Hierarchical Attention Transformer, which segments long text without sticking to the document structure, and Longformer, which uses sparse attention.

**Questions To Authors:**

* Page 9, Efficiency Analysis `12-layer models models` -> `12-layer models`
* Figure 6 and the corresponding text description is Section 4.1 is difficult to understand. Also it is not very clear how these experiments relate to the paper's contribution.
* Figure 1 (b): is this training loss or loss calculated on a hold-out dev set?
* Not sure whether Figure 1 (a) and Figure 2 are duplicated, it seems Figure 2 is a more detailed version or version with example?
* Suggest adding full name of SRAM when you first mention it.

**Reasons To Accept:**

* The proposed architecture sounds promising: the hierarchical transformers have been investigated a lot for different NLP tasks; the paper furthers this direction by developing a new sparse attention kernel which can be used as a drop-in replacement of existing standard attention.
* The authors evaluate the proposed transformer variant on various tasks to assess the effectiveness of the learned representations. The reported results surpass those of both HAT and Longformer.
* The authors categorize the literature on long-document transformers into two categories and provide a nice overview.

**Reasons To Reject:**

* Although the proposed architecture outperforms baselines (i.e., LED base) of similar scale, there is actually a big gap between the reported results and SOTA results (even some of them are really LLMs, e.g., BART-large SLED) on Scrolls, probably on other datasets as well.
* Considering there are Long-context LLMs (e.g., million tokens) available, the maximum sequence length (8192 for pre-training) considered in this paper may not be very exiting.

---

> ### Author Rebuttal · Authors · 2024-05-31
>
> **Although the proposed architecture outperforms baselines (i.e., LED base) of similar scale, there is actually a big gap between the reported results and SOTA results (even some of them are really LLMs, e.g., BART-large SLED) on Scrolls, probably on other datasets as well.**
>
> We, too, are very interested in scaling our ideas to larger model sizes and task sets, such as code which is also structured data. We consider our work as a proof-of-concept to demonstrate that incorporating natural hierarchical structure via sparse attention is possible.
>
> **Considering there are Long-context LLMs (e.g., million tokens) available, the maximum sequence length (8192 for pre-training) considered in this paper may not be very exciting.**
>
> Thanks to our hierarchical attention design and the implementation of our kernel, we are able to process 8192 tokens as input during training and inference using standard (academic) compute hardware. While recently LLMs have been extended to larger contexts, they do not consider the input structure and require large distributed compute (eg, RingAttention) for training or alternative non-structured memory mechanisms.
>
> Training/Fine-tuning our pre-trained structured model with longer inputs is possible both theoretically and computationally. Figure 8 shows how our hierarchical model scales to larger inputs in terms of runtime. In preliminary experiments, we increased the input length to 30k for NarrativeQA (Nrtv) in Table 4, resulting in slight improvements from 14.2 -> 15.3. However, most datasets that we evaluate on do not require input lengths larger than 8k.
>
> **Figure 1 (b): is this training loss or loss calculated on a hold-out dev set?**
>
> In Figure 1(b) we report the training loss as also done in prior work (eg, CRAMMING) for evaluating pre-training. However, the validation loss behaves similarly, and we will also report it in the final paper.

---

> > ### Comment · Reviewer_vjsH · 2024-06-03
> >
> > Just acknowledging I have read the response; I keep my original rating.

---

### Official Review · Reviewer_XCB9 · 2024-05-13

**Rating:** 7
**Confidence:** 3
**Ethics Flag:** 2

**Summary:**

This paper presents an approach to explicitly encode document structure inductive biases in the Transformer architecture's attention mechanism. The approach is interesting and well executed however the motivation for introducing the proposed approach is not clear besides "more efficient" and "sparse". A variety of techniques have been proposed to induce sparsity in the attention mechanism albeit tailored to specific tasks. This paper claims to improve general "long document tasks". To satisfy this claim more experiments should be included on a variety of tasks.

**Reasons To Accept:**

Clarity, well written. Proposes a novel, well executed and detailed approach.

**Reasons To Reject:**

The motivation for encoding structure explicitly is not very clear and a variety of other techniques have been proposed for inducing sparsity. The code should be included.

---

> ### Author Rebuttal · Authors · 2024-05-31
>
> **The motivation for encoding structure explicitly is not very clear and a variety of other techniques have been proposed for inducing sparsity. The code should be included.**
>
> The main goal of this work was to design and implement an efficient Transformer for long documents using sparse attention. In this process, the choice of the attention pattern is a central question. As has been shown in studies with both humans [1,2] and language models [3], the document structure can be a useful signal when solving tasks on long documents, document structure offers a natural sparse attention pattern to process long documents more efficiently. To fully exploit document structure in sparse attention, a major challenge is the implementation of an attention kernel that handles arbitrary attention patterns efficiently. We solve this challenge and present results that show promising performance. As stated in the introduction, we will release our code and models.
>
> **References**
>
> [1] Barbara M. Taylor and Richard W. Beach. The Effects of Text Structure Instruction on Middle-Grade Students’ Comprehension and Production of Expository Text. Reading Research Quarterly, 19(2):134–146, 1984.
>
> [2] John T. Guthrie, Tracy Britten, and K. Georgene Barker. Roles of Document Structure, Cognitive Strategy, and Awareness in Searching for Information. Reading Research Quarterly, 26(3), 1991.
>
> [3] Jan Buchmann, Max Eichler, Jan-Micha Bodensohn, Ilia Kuznetsov, and Iryna Gurevych.
> Document structure in long document transformers. In Conference of the European Chapter
> of the Association for Computational Linguistics (EACL), 2024.

---

> > ### Comment · Reviewer_XCB9 · 2024-06-04
> >
> > I've read the response, I keep my original rating.

---

### Official Review · Reviewer_P6rv · 2024-05-14

**Rating:** 7
**Confidence:** 5
**Ethics Flag:** 1

**Summary:**

The authors present a novel Transformer architecture, namely Hierarchical Document Transformer (HDT), targeting long document processing, which explicitly imposes document structure as an inductive bias in the attention mechanism using special anchor tokens for different document levels (document, sections, and sentences) in the hierarchy. The authors also designed a custom hierarchical attention kernel, influenced by FlashAttention (Dao et al., 2022-2023) to improve efficiency (clock time and memory). They experiment with both Encoder-Only and Encoder-Decoder models comparing to notable baselines, Longformer and HAT for Encoder-Only, and LED for Encoder-Decoder models. The authors showcase that the newly proposed architecture outperforms the baselines on average considering results on SciRepeval proximity (similarity) tasks for Encoder-Only models, and FacetSum and SCROLLs for Encoder-Decoder models. In their efficiency analysis, the authors find their models to be faster in terms of clock time and TFLOPs and more memory-efficient compared to Longformer, on par with HAT.

**Questions To Authors:**

* Why you did not consider expanding the pretraining corpora beyond Arxiv and Patents? From a practical perspective, i.e., people reusing your model for real-life scenarios, this limits the model to fewer domains, while from a benchmarking perspective, it may give your model an extra edge (Arxiv -> SciRepeval) compared to HAT, which was not pre-trained from scratch. I would consider expanding the pertaining corpus with long documents, like those in legal corpora, e.g., the English part of the MultiLegalPile, or general documents >1K by filtering ThePile or C4. Wikipedia documents are also relatively short, close to 200 tokens on average.
* I would like to see a comparison between HDT, HAT, and Longformer on long-form ListOps with Ks of numbers and ops. I think it would be a more interesting comparison instead of BERT which completely lacks any form of anchor tokens or explicit hierarchical attention.
* What are the scores for "Feeds-M" and "High. Infl." refer to in Table 2?

**Reasons To Accept:**

* The newly proposed method, HDT, is a great intuitive step forward for Hierarchical Transformers, similar to HAT, that try to impose document structure in the modeling.
* The newly proposed method, HDT, seems to outperform several notable baselines in different benchmarks.
* The paper is nicely written, organized, and easy to follow with great figures that can help readers understand the technical issue of processing long documents and the newly proposed architecture.

**Reasons To Reject:**

* The benchmarking is mostly limited to document proximity (similarity) tasks and summarization. Long document classification with datasets such as ECtHR, MIMIC, CONTRACT-NLI, or QuALITY,  is not part of the evaluation.
* The evaluation is mainly focused on the size regime of medium-sized models (~100M params.), which leaves open the question of how HDT would benefit from scaling to larger models with Bs of params, where it could also have a more fair comparison to LongT5 and CoLT5 models.
* The authors did not consider the idea of warm-starting (model-recycling) used by Longformer, HAT, and LongT5, which is another dimension of efficiency concerning saving compute time and resources.

---

> ### Author Rebuttal · Authors · 2024-05-31
>
> **Long document classification**
>
> During the rebuttal period, we did additional experiments on the ECtHR-LJP dataset, a multi-label document classification task used in the HAT paper. We trained both models using our implementation and evaluated them with the micro-F1 metric. Our model shows marginal improvements compared to HAT (79.1 -> 79.2). Note that results for CONTRACT-NLI (CNLI) and QuALITY (QALT) can be found in Table 4, evaluated as a text-to-text generation task instead of a classification task. We have also applied for access to the MIMIC dataset and will add results once we have been approved.
>
> **Scaling to larger models with Bs of params**
>
> The main focus of our paper is on processing structured datasets which are currently smaller than unstructured datasets used for training LLMs. However, we agree that scaling the model and datasets (eg, using additional structured data or combining structured data with unstructured data) is an interesting avenue which we want to explore in future work.
>
> **Warm-starting (model-recycling)**
>
> This is indeed interesting, and we did some preliminary experiments on this. We found that in contrast to other models like Longformer and LongT5, our model can be stably trained from scratch which we consider an advantage of our model. One difficulty in warm-starting HDT from existing pre-trained models is that we use additional anchor tokens which must be trained as well. We plan to investigate this in the future.
>
> **Pre-training with ArXiv data**
>
> Thanks for pointing this out. The reason that we choose SciRepEval as benchmark and Arxiv as a part of our pre-training data is that both include structured data. However, note that Arxiv documents make up for only 15% of our pre-training data. For this rebuttal, we conducted additional experiments and pre-train our model from scratch without ArXiv documents, following the same setting as before (1 GPU day), and evaluating it on SciRepEval. We find that with contrastive fine-tuning, our model maintains its performance (70.55 -> 70.33), while the pretrained-only model performance drops slightly (66.27 -> 65.16), yet still outperforms all baselines.
>
> **Comparison between HDT, HAT and Longformer on ListOps**
>
> We have conducted new experiments comparing to Longformer and HAT on sequences up to 512 tokens. HDT (0.794) significantly outperforms BERT (0.599), Longformer (0.622) and HAT (0.584) with 5 operands and a maximum tree depth of 20.
>
> **Scores in Table 2**
>
> See caption.

---

> > ### Comment · Reviewer_P6rv · 2024-06-06
> >
> > Thanks for your thorough response. I've read the response. I think it will be great to include all additional experiments in the final version of your article, including results with MIMIC, if possible. I keep my original rating.

---

### Official Review · Reviewer_9hSs · 2024-05-17

**Rating:** 6
**Confidence:** 4
**Ethics Flag:** 1

**Summary:**

This paper studies hierarchical document representation using different position representations for the hierarchy of the document as section, paragraph and token; the attention mask is adjusted accordingly with a more sparse representation of the whole document. The proposed transformer architecture is compared with existing transformer architectures and some sparse attention transformers approaches as only encoder and encoder-decoder transformers  improves over mathematical reasoning and language tasks.

**Questions To Authors:**

- In table 5, the bold values do not always correspond to the best value, e.g. time 77.84 for HAT is smaller than 79.8 for HDT-E. What is the rationale behind using bold values?
- Could you please add the missing description of notation used in equation 2-7?

**Reasons To Accept:**

- The usage of position information for different hierarchy is interesting.
- The results on SciRepEval Proximity tasks, and it leads to comparable or better results on SCROLLS summarization, QA, and NLE benchmark

**Reasons To Reject:**

-The experiments do not compare to more recent approaches as Fast Attention Over Long Sequences With Dynamic Sparse Flash Attention, https://openreview.net/pdf?id=UINHuKeWUa
-The writing up of methodology could be improved:
The description of HPE and hierarchical attention are not very clear. The same index $i$ is used for position and mask, superscript $1$,$2$,$3$ used in Equation 4,5 & 6 are not explained. Fig 10 is referenced for a mask that is sparse in practice, but it’s not clear what is the document structure for that example? Providing the text next to it could help to support this claim as currently it looks like a toy example.

---

> ### Author Rebuttal · Authors · 2024-05-31
>
> **The experiments do not compare to more recent approaches as Fast Attention Over Long Sequences With Dynamic Sparse Flash Attention**
>
> Thanks for the suggestion. We first want to remark that the suggested paper considers auto-regressive decoder-only models and hence cannot be compared to encoder models directly. However, for this rebuttal, we adapted their model to an encoder-only variant by dropping the causal mask and using MLM for pre-training. However, we found this model difficult to train and could not get sensible results within the rebuttal period. We will continue experimenting with this model after the rebuttal and report our results in the paper, if successful.
>
> **The writing up of methodology could be improved: The description of HPE and hierarchical attention are not very clear. The same index $i$ is used for position and mask, superscript 1, 2, 3 used in Equation 4,5 & 6 are not explained.**
>
> The superscript $l=1,2,3$ consistently denotes the level of hierarchy and is explained in Figure 3. Both indices $i, j$ consistently denote positions. We will make this more clear at the start of the methodology section and make sure that all symbols and indices are named in the main text as well.
>
> **Fig 10 is referenced for a mask that is sparse in practice, but it’s not clear what is the document structure for that example? Providing the text next to it could help to support this claim as currently it looks like a toy example.**
>
> Thank you. The examples we use actually are real documents from arXiv (we will clarify this further) and show the first 1k tokens of each document (~25%) as described in the caption. The actual text is indeed shown at the axes but unfortunately not legible due to the figure resolution to keep the pdf size reasonable. We will provide high-resolution versions of these figures on our project page.
>
> **In table 5, the bold values do not always correspond to the best value, e.g. time 77.84 for HAT is smaller than 79.8 for HDT-E. What is the rationale behind using bold values?**
>
> Thank you for pointing this out. This was indeed a typo, which will be corrected. Our method is slightly (2 ms) slower than HAT, which is due to our dynamic sample-specific attention pattern while HAT uses the same attention pattern for every sample.
>
> **Could you please add the missing description of notation used in equation 2-7?**
>
> We will make sure that all symbols and indices are described in the main text.

---

> > ### Comment · Reviewer_9hSs · 2024-06-06
> >
> > Thanks for the response. I believe with the changes will improve the final. I keep my original rating.

---

### Decision · Program_Chairs · 2024-07-10

**Decision:**

Accept

**Comment:**

Paper provides a good demonstration of a technique to improve efficiency (sample, and computational) to the increasingly important problem of long-context modeling, based on document structured and operationalized via sparse attention. Experimental work verifies the approach on realistic workloads. Although very long contexts are not used in the paper, this is a comprehensive demonstration that structured attention based on document properties holds promise.

Paper is relatively clearly written, and reviewers and rebuttal comments indicated it will be of interest to many readers.